# New-Onset and Relapsed Kidney Histopathology Following COVID-19 Vaccination: A Systematic Review

**DOI:** 10.3390/vaccines9111252

**Published:** 2021-10-29

**Authors:** Henry H. L. Wu, Philip A. Kalra, Rajkumar Chinnadurai

**Affiliations:** 1Department of Renal Medicine, Lancashire Teaching Hospitals NHS Foundation Trust, Preston PR2 9HT, UK; 2Faculty of Biology, Medicine and Health, University of Manchester, Manchester M13 9PL, UK; Philip.Kalra@nca.nhs.uk (P.A.K.); Rajkumar.Chinnadurai@nca.nhs.uk (R.C.); 3Department of Renal Medicine, Salford Royal Hospital, Northern Care Alliance Foundation Trust, Salford M6 8HD, UK

**Keywords:** COVID-19 vaccination, new-onset, relapse, kidney disease, histopathology, systematic review, COVID-19, SARS-CoV-2

## Abstract

Introduction: The introduction of COVID-19 vaccination programs has become an integral part of the major strategy to reduce COVID-19 numbers worldwide. New-onset and relapsed kidney histopathology have been reported following COVID-19 vaccination, sparking debate on whether there are causal associations. How these vaccines achieve an immune response to COVID-19 and the mechanism that this triggers kidney pathology remains unestablished. We describe the results of a systematic review for new-onset and relapsed kidney histopathology following COVID-19 vaccination. Methods: A systematic literature search of published data up until 31 August 2021 was completed through the Preferred Reporting Items for Systematic Reviews and Meta Analyses (PRISMA) guideline. Research articles reporting new onset or relapsed kidney histopathology in adult patients (>18 years) following COVID-19 vaccination were included for qualitative review. Only full-text articles published in the English language were selected for review. Results: Forty-eight cases from thirty-six articles were included in the qualitative synthesis of this systematic review. Minimal change disease (19 cases) was the most frequent pathology observed, followed by IgA nephropathy (14 cases) and vasculitis (10 cases). Other cases include relapse of membranous nephropathy, acute rejection of kidney transplant, relapse of IgG4 nephritis, new-onset renal thrombotic microangiopathy, and scleroderma renal crisis following COVID-19 vaccination. There was no mortality reported in any of the included cases. Patients in all but one case largely recovered and did not require long-term renal replacement therapy. Conclusion: This systematic review provides insight into the relationship between various kidney pathologies that may have followed COVID-19 vaccination. Despite these reported cases, the protective benefits offered by COVID-19 vaccination far outweigh its risks. It would be recommended to consider early biopsy to identify histopathology amongst patients presenting with symptoms relating to new-onset kidney disease following vaccination and to monitor symptoms for those with potential relapsed disease.

## 1. Introduction

The vulnerability of patients with kidney disease to Coronavirus Disease 2019 (COVID-19) have been well-established since the onset of this ongoing pandemic [1,2,3]. Given that there are no curative solutions currently available to treat or prevent COVID-19 manifestations and mortality, the introduction of vaccination programs following the discovery of COVID-19 vaccines, in addition to public health infection mitigation measures, are the main strategies to reduce COVID-19 numbers [4]. Patients with kidney disease should avoid live-replicating microbial-vectored vaccines due to their likelihood of having a compromised immune system compared to the general population without kidney disease [5]. Replication-defective viral-vectored and messenger RNA (mRNA) vaccines such as the BNT162b2 (Pfizer-BioNTech, Cambridge, MA, USA), ChAdOx1 nCoV-19 (Oxford-AstraZeneca, Oxford, UK), mRNA-1273 (Moderna, Cambridge, MA, USA), Ad26.COV2.S (Janssen, Beerse, Belgium), PiCoVacc (Sinovac, Beijing, China), Gam-COVID-Vac (Sputnik V, Moscow, Russia), Covaxin BBV152 (Bharat Biotech, Hyderabad, India), Convidecia AD5-nCOV (CanSino, London, UK), and BBIBP-CorV (SinoPharm, Beijing, China) vaccines, have been determined as safe and approved for use in an accelerated manner since early 2020. Individuals receiving viral-vectored vaccines such as the Oxford-AstraZeneca can generate a T-cell response, a cluster of differentiation between 8+ and 4+ (CD8+ and CD4+) expansion, and a type 1 T-helper cell-biased (Th1-biased) response with the production of interferon-c (IFN-c), tumor necrosis factor-alpha (TNF-α), interleukin-2 (IL-2), and immunoglobulin G1 and G3 (IgG1 and IgG3) subclass antibodies. Additionally, there is a pioneer mechanism of action in mRNA vaccines, such as in Pfizer-BioNTech and Moderna. Lipid nanoparticle nucleoside-modified mRNA encodes the SARS-CoV-2 spike protein, mediating host attachment and SARS-CoV-2 viral entry [6,7,8].

The mechanisms of how these vaccines achieve an immune response to COVID-19 have been reported to associate with newfound or relapse of podocytopathy, glomerular disease, and other intrarenal pathologies. As of 31 August 2021, there have been no systematic reviews published to summarize findings from COVID-19 vaccine-induced kidney disease. Here, we provide a systematic clinical review ofx the current literature to delineate the range of kidney histopathologies that were elicited following COVID-19 vaccination.

## 2. Materials and Methods

### 2.1. Eligibility Criteria

All research articles reporting new onset or relapsed kidney histopathology in adult patients (>18 years) following COVID-19 vaccination were included. These involved pathologies in both native and transplanted kidneys. We only selected full-text articles published in the English language. Only studies published before 31 August 2021 were included in this review.

### 2.2. Search Strategy and Study Selection

A systematic literature search was conducted by two independent authors (H.W. and R.C.) in the following database: ‘PubMed’, ‘Web of Science’, ‘EMBASE’, and ‘Medline-ProQuest’. The search terms incorporated the following: ‘COVID-19 Vaccination’ AND ‘Kidney Histopathology’; ‘COVID-19 Vaccination’ AND ‘Renal Histopathology’; ‘SARS CoV-2 Vaccination’ AND ‘Kidney Histopathology’; ‘SARS CoV-2 Vaccination’ AND ‘Renal Histopathology’; ‘Pfizer-BioNTech Vaccine’ AND ‘Kidney Histopathology’; ‘‘Pfizer-BioNTech Vaccine’ AND ‘Renal Histopathology’; ‘Moderna Vaccine’ AND ‘Kidney Histopathology’; ‘Moderna Vaccine’ AND ‘Renal Histopathology’; ‘Oxford-AstraZeneca Vaccine’ AND ‘Kidney Histopathology’; ‘Oxford-AstraZeneca Vaccine’ AND ‘Renal Histopathology’; ‘Sinovac Vaccine’ AND ‘Kidney Histopathology’; ‘Sinovac Vaccine’ AND ‘Renal Histopathology’; ‘Janssen Vaccine’ AND ‘Kidney Histopathology’; ‘Janssen Vaccine’ AND ‘Renal Histopathology’; ‘Sputnik V Vaccine’ AND ‘Kidney Histopathology’; ‘Sputnik V Vaccine’ AND ‘Renal Histopathology’; ‘Bharat Biotech Vaccine’ AND ‘Kidney Histopathology’; ‘Bharat Biotech Vaccine’ AND ‘Renal Histopathology’; ‘CanSino Vaccine’ AND ‘Kidney Histopathology’; ‘CanSino Vaccine’ AND ‘Renal Histopathology’; ‘Sinopharm Vaccine’ AND ‘Kidney Histopathology’’; and ‘Sinopharm Vaccine’ AND ‘Renal Histopathology’. The articles were screened by H.W. and R.C. for relevance and duplicate publications were removed. Duplicate screening and the eligibility check was performed by both H.W. and R.C. The study selection process was carried out using the Preferred Reporting Items for Systematic Reviews and Meta Analyses (PRISMA) guideline (Figure 1).

### 2.3. Data Extraction

Data including patient demographics (age and sex), time to presentation from the day of previous vaccination, comorbidities, brand of vaccine administered, number of vaccine doses given, kidney parameters at baseline and at time of presentation, treatment received following diagnosis, and clinical outcome were extracted from each included article. Data are described in the results section of this article and presented in tabular form.

### 2.4. Study Registration

A pre-defined review protocol was registered at the PROSPERO international prospective registry of systematic reviews under registration number CRD42021275949.

## 3. Results

A total of 1142 articles were identified on an initial search. After exclusion of duplicates and articles that did not fulfill the study inclusion criteria, thirty-six articles were included in the qualitative synthesis of this systematic review. The reports of forty-eight cases identified from these articles are presented by groups based on histopathology.

### 3.1. Minimal Change Disease

Minimal change disease (MCD) was the most common histopathology reported following COVID-19 vaccination (eleven new onset and eight relapsed cases), with nephrotic syndrome as the common presentation in these cases (see Table 1).

A total of eleven cases presenting with de novo nephrotic syndrome following COVID-19 vaccination were diagnosed with MCD on kidney histopathology [9,10,11,12,13,14,15,16,17,18,19]. The median interquartile range (IQR) age of this group was 63 (50 to 75) years, with a male predominance (7:4). The median (IQR) time between the COVID-19 vaccination and time of presentation was 7 (7 to 13) days. Six of these eleven cases (four after the first dose and two after the second dose) were reported following Pfizer-BioNTech vaccination. Five patients had concurrent acute kidney injury with two eventually needing renal replacement therapy. All cases (except one in which the management regime was not reported) received high-dose steroid treatment and all patients showed resolution of proteinuria and acute kidney injury (AKI). As cited by the majority of these case reports, the possible trigger for podocytopathy was a T-cell-medicated immune response to viral mRNA.

There were eight relapsed MCD cases reported following COVID-19 vaccination in patients who had a previous histopathological diagnosis of MCD [13,20,21,22,23,24]. The median (IQR) age of this group was 34 (31 to 40) years, with an equal distribution between males and females. The median (IQR) time of presentation following COVID-19 vaccination was 7 (2 to 20) days. Five of the eight relapsed MCD cases (three after the first dose and two after the second dose) occurred following administration of the Pfizer-BioNTech vaccine. AKI was reported in one case. All eight patients achieved resolution of proteinuria and recovery following treatment with high-dose steroids. 

### 3.2. IgA Nephropathy

IgA nephropathy was the second most-common histopathological diagnosis reported (six new onset and six previously known cases). Macroscopic hematuria was observed in eight of the twelve cases. Two patients presenting with macroscopic hematuria following COVID-19 vaccination who did not have a kidney biopsy were presumed to have IgA nephropathy (see Table 2).

Six cases of new onset IgA nephropathy were reported in patients presenting with macroscopic hematuria following COVID-19 vaccination [25,26,27,28,29]. The median (IQR) age of the group was 40 (27 to 50) years, with a male: female ratio of 4:2. The median (IQR) time between the onset of symptoms and COVID-19 vaccination was just 1 (1 to 2) day. Five of the cases were reported following the second dose of the Moderna vaccine and the other following a second dose of the Pfizer-BioNTech vaccination. Five patients presented with sub-nephrotic proteinuria. AKI was reported in two patients who had crescentic IgA nephropathy reported in the histopathology and both were managed with high-dose steroids and cyclophosphamide. The other four patients received supportive management with eventual resolution of hematuria. 

Macroscopic hematuria was reported following COVID-19 vaccination in six cases with a previously known histopathological diagnosis of IgA nephropathy [30,31,32]. The group had a median (IQR) age of 38 (25 to 48) years and female predominance (5:1), with a median (IQR) time between COVID-19 vaccination and onset of symptoms of 2 (1 to 2) days. In half of the cases, patients received the Pfizer-BioNTech vaccine (one developed symptoms after the first dose and two after the second dose), while the other half received the Moderna vaccine (all three after the second dose). Three of the six patients had sub-nephrotic proteinuria. Management for all six patients was supportive and a resolution of symptoms was noted in all the reported cases.

Park et al. [29] reported two female patients, aged 22 and 39 years, presenting with gross hematuria and mild proteinuria without a rise in creatinine within two days of the second dose of Moderna vaccination, with spontaneous resolution noted on follow-up at 1 month. One patient had a history of receiving episodic steroids for IgA vasculitis at the age of 10 years old. Both patients did not have a kidney biopsy and were presumed to have IgA nephropathy following COVID-19 vaccination.

### 3.3. Vasculitis

Vasculitis was the third most-common histopathology (ten cases) reported following COVID-19 vaccination from our review (see Table 3). The temporal associations in these reported cases generate the hypothesis of immune-mediated diseases triggered by COVID-19 vaccinations, although the mechanisms remain unclear.

A total of four cases of ANCA-associated vasculitis have been reported, including two cases of MPO-ANCA-associated vasculitis following the second dose of the Pfizer-BioNTech vaccination and two cases of PR3-ANCA-associated vasculitis following Moderna vaccination (one after the first dose and one after the second dose) [26,33,34,35]. The clinical presentation was AKI and hematoproteinuria in all four cases. The time interval between vaccination and presentation varied widely between the cases (from 14 to 49 days). In each of the cases, kidney biopsy revealed crescentic glomerulonephritis and all four patients received treatment with standard immunosuppression. One patient remained dialysis-dependent treatment, while the other patients had recovered kidney function at the time of reporting.

Gillion et al. [36] reported the case of a 77-year-old man who presented as feeling unwell four weeks after receiving the first dose of the AstraZeneca vaccine and his routine blood test suggested AKI. There was no hematoproteinuria and his ANCA serology was negative. The kidney biopsy showed diffuse interstitial edema with non-caseating non-necrotizing granulomas around small vessels, with one small vessel displaying fibrinoid necrosis. His positron emission tomography scan showed findings suggestive of vasculitis. There were no clinical or radiological features of sarcoidosis. He responded well to steroid treatment, with complete resolution of AKI. With all other tests for granulomatous conditions being negative, this was thought to be possibly linked with COVID-19 vaccination.

Obeid et al. [37] reported a case of an IgA vasculitis flare in a 78-year-old lady with previous renal and gastrointestinal (GI) involvement who had been in remission for two years without any immunosuppression. She presented with GI symptoms of diarrhea and abdominal pain a week after the first dose of the Moderna vaccination. She also developed palpable purpura in the hips and lower limbs. Renal manifestations included macrohematuria, sub-nephrotic proteinuria, and a mild rise in serum creatinine. Her symptoms resolved with intravenous steroids. A serum antinuclear antibody screening test on fixed Hep-2 cells showed the autoreactivity of the patient’s IgA after the mRNA vaccination, which was not present before the vaccination, supporting the link between vaccination and IgA vasculitis flare. Park et al. [29] reported another case of IgA vasculitis in a 67-year-old man who presented with gross hematuria and a lower extremity rash one month after his first dose of the Moderna vaccine. A skin biopsy showed IgA vasculitis. The second vaccine dose worsened the skin rash but did not affect the renal symptoms. Both renal and skin symptoms resolved with a week course of oral prednisolone 40 mg daily. 

Two cases of anti-GBM disease have been reported following COVID-19 vaccination [25,38]. One patient presented a day after the second dose of the Pfizer-BioNTech vaccination with macroscopic hematuria and the other presented two weeks after the second dose of the Moderna vaccination with systemic illness (fever, anorexia, and nausea) and gross hematuria. In both cases, the immunology screening showed positive anti-GBM titers and the kidney histopathology demonstrated linear IgG-staining of the GBM as well as active cellular crescents. Both of these patients developed AKI and were treated with pulsed methylprednisolone, cyclophosphamide, and plasma exchange. One patient remained dialysis-dependent at the time of report, while the clinical outcome of the other patient was not reported.

Tuschen et al. [39] reported a patient with a flare up of lupus nephritis a week following the first dose of the Pfizer-BioNTech vaccination. She was a known case of class V lupus nephritis in remission who was receiving hydroxychloroquine maintenance treatment. The patient developed nephrotic syndrome following vaccination and a repeat kidney biopsy showed features of class V and II lupus nephritis with slight focal and segmental mesangial hypercellularity, granular immunoreactivity for immunoglobulin G, and complement 3c along the glomerular capillary walls and within the mesangium. She was started on mycophenolate and high-dose oral prednisolone but her proteinuria was slow to resolve. 

### 3.4. Other Cases

Aydin et al. [40] reported the relapse of membranous nephropathy (MN) in a 66-year-old female following the Sinovac vaccine, with development of nephrotic syndrome. Secondary causes of MN such as malignancy, infection, and medication-induced MN were excluded, which led to speculation regarding whether mechanisms of immune system dysregulation following vaccination triggered this presentation. 

Del Bello et al. [41] reported a case of acute rejection in a 23-year-old female with a kidney transplant, which occurred following the second dose of the Pfizer-BioNTech vaccine. The patient underwent deceased donor kidney transplantation for nephronophthisis 18 months earlier and post-transplant clinical progress was uneventful in terms of immunosuppressive therapy (she was receiving tacrolimus, mycophenolic acid, and low-dose steroid). 

Masset et al. [42] reported a relapse of immunoglobulin G4-related (IgG4-RD) nephritis following administration of the Pfizer-BioNTech vaccine in a 66-year-old male patient. It was postulated that the relapse of IgG4-RD nephritis occurred due to direct immune activation following vaccination, a process of chronic immune activation following pauci-symptomatic allergic reaction, or a combination of these two mechanisms. 

A 35-year-old previously healthy male patient presenting with nephrotic-range proteinuria and microscopic hematuria following the first dose of the Pfizer-BioNTech vaccine was reported by De Fabritiis et al. [43], with the final histopathological diagnosis found to be renal thrombotic microangiopathy (renal TMA). The patient had a positive severe acute respiratory syndrome coronavirus 2 polymerase chain reaction (SARS-CoV-2 PCR) test seven days after receiving his vaccine prior to the onset of urinary abnormalities. The temporal relationship between these events emphasizes the likelihood that complexes of COVID-19-specific antibodies and SARS-CoV-2 viral antigens elicited endothelial injury in combination, leading to renal TMA. Fortunately, the patient did not develop kidney failure and he eventually achieved complete remission of proteinuria and microscopic hematuria with steroid treatment. 

Oniszczuk et al. [44] described the case of a 34-year-old female patient who presented with severe hypertension and AKI one week following her first dose of the Pfizer-BioNTech vaccine. Physical examination on admission revealed thickened skin on the face and sclerodactyly at the back of both hands and wrists, as well as oral telangiectasia. Further investigation findings followed by a kidney biopsy (which revealed two globally sclerosed glomeruli with secondary ischemic glomerular changes) were suggestive of a scleroderma renal crisis. The patient was discharged after one week of hospitalization and following commencement of antihypertensive treatment, with a stable kidney function and normalized blood pressure.

## 4. Discussion

A considerable range of kidney histopathologies were observed following COVID-19 vaccination. As the dominant pathology reported in our review, MCD is defined as diffuse losses of the visceral epithelial cell-foot processes in the kidney glomeruli, resulting in podocyte effacement, vacuolation, and growth of microvilli on the visceral epithelial cells, which leads to excess protein losses in urine [45]. The pathogenesis remains unestablished in regard to how COVID-19 vaccination induces MCD and nephrotic syndrome, although dysregulation of T-cell-mediated immunity is widely speculated to be the main cause. Increased formation of the permeability factor from enhanced type 2 T-helper cell activity causing cytokine release has been hypothesized as the pathogenesis of MCD [46]. Administration of influenza, hepatitis B, pneumococcal, and measles to tetanus–diphtheria–poliomyelitis vaccines have previously been reported to be associated with the onset of MCD [47]. Other contributing immune-related triggers include allergic reactions, bee stings, malignancy, autoimmune disease, and medications such as D-penicillamine [45,48].

IgA nephropathy is the most common primary cause of glomerulonephritis [49]. It is an immune-complex pathology characterized by deposition of mesangial immunoglobulin A1 (IgA1) with or without concurrent immunoglobulin G (IgG) and complement 3 (C3). Poorly galactosylated IgA1 forms immune complexes in the mesangium. Glycan-specific IgA or IgG targets these poorly galactosylated regions, leading to impaired liver removal of these complexes and their increased affinity for mesangial cells [50,51]. IgA nephropathy is a condition with a multi-hit causative mechanism involving genetically predisposed variants encoding galactosylation and environmental triggers such as infection, environmental chemical exposure, and dietary imbalances that lead to increased anti-glycan IgG and IgA production [52]. IgA nephropathy is not frequently reported following other (non-COVID) vaccinations, although previous cases associated with the influenza vaccine have been well-documented in cases involving both native and transplanted kidneys [47]. Van den Wall Bake et al. [53] observed that the intramuscular inactivated influenza vaccine elicited hyperresponsiveness in a cohort of patients with existing IgA nephropathy, with excessive production of IgA1 monomers. An explanation for the associations between COVID-19 vaccination and IgA nephropathy is not fully established. One explanation is that there may be greater production of anti-glycan antibodies which cross-react with poorly galactosylated IgA1, since mucosal immune responses are not stimulated following COVID-19 vaccination. Furthermore, increased antibody production is expected for patients receiving mRNA vaccines, which induces more robust T-helper cell and B-cell responses in the germinal center. Another observation relates to a spike in IgA production in healthy individuals following mRNA vaccines, similar to what was reported for influenza vaccines [54]. Scenarios where a subclinical IgA nephropathy becomes clinically apparent following COVID-19 vaccination is another possibility. 

It is of interest to note that from our review, the majority of MCD cases were reported following the first dose of the COVID-19 vaccination, while most IgA nephropathy cases were reported following the second dose of vaccination. In addition, the median time for symptom onset in IgA nephropathy cases (1 day) was much shorter than that of MCD (7 days). Both of these observations are in support of the hypothesis for the generation of T-cell-mediated injury in MCD and antibody-mediated immune response in IgA nephropathy; the latter fits precisely with the synpharyngitic relapse of IgA nephropathy that is recognized in normal clinical practice. 

The relationship between vasculitic disease and COVID-19 vaccination remains poorly elucidated. No direct proof exists to suggest a connection between vaccine response and development of ANCA vasculitis. Jeffs et al. [55] noted an increased production of ANCA following viral RNA influenza and rabies vaccines. Once the study subjects were treated with ribonuclease, it was observed that there was a significantly reduced ANCA response to RNA vaccines. This encourages discussion as to whether there is a direct relationship between ANCA vasculitis and reactions to RNA. Further real-world study is required to establish whether ANCA vasculitis more frequently occurs following the use of mRNA vaccines versus other types. The development of IgA vasculitis and anti-GBM disease following COVID-19 vaccination has been attributed to a hyperactive immune response towards SARS-CoV-2 [37,38,56]. Although there were initial suggestions of a causative mechanism resulting from COVID-19 vaccination, further reports of granulomatous vasculitis, lupus nephritis, membranous nephropathy, acute transplant rejection, IgG4-related nephritis, renal TMA, and scleroderma renal crisis will be required to warrant greater validity in recognizing these associations.

Some publications that reported kidney disease following COVID-19 vaccination were not included in our review, as these reports did not fulfill study inclusion criteria. In an article published in the Spanish language, De la Flor et al. [57] recently reported the first case of acute interstitial nephritis following administration of the Pfizer-BioNTech vaccine. More cases are required to validate any existing association between the onset of tubulointerstitial kidney disease and COVID-19 vaccination. Hanna et al. [58] reported IgA nephropathy presenting as macroscopic hematuria in two pediatric patients (13 and 17 years old, respectively) who received the Pfizer-BioNTech vaccine in a case series from the United States (US). It was not recommended for children to receive COVID-19 vaccination until May 2021 when the US Food and Drug Administration (FDA) granted emergency use authorization to include all children aged 12 to 15 years for the Pfizer-BioNTech vaccine (emergency use authorization for the Pfizer–BioNTech vaccine for individuals aged ≥ 16 years was only granted on Dec 2020 by the FDA). These cases support the need for a stricter monitoring of symptoms following COVID-19 vaccination in pediatric patients who may have undiagnosed kidney disease previously. 

From the forty-eight cases that were evaluated in our review, mRNA vaccines (Pfizer-BioNTech (23 cases) and Moderna (17 cases)) appear to be the most frequent COVID-19 vaccines associated with post-vaccination kidney disease development (see Figure 2). The higher number of cases can be attributed to the immune response generated to the mRNA vaccines as discussed above or probably due to the fact that the vast majority of cases were reported from a select number of countries across North America, Europe, and Asia, where Pfizer-BioNTech and Moderna vaccines have been more accessible and commonly available in established vaccination programs. Furthermore, our systematic review reported cases where there is a defined kidney histopathology following COVID-19 vaccination. We acknowledge that the real frequency of acute kidney impairment following COVID-19 vaccination as a result of inflammatory complications may be underreported in literature. 

## 5. Conclusions

Our review of the reported cases provides insight into the likely immune-mediated relationship between various kidney pathologies developing following COVID-19 vaccination. The number of reported cases is relatively small in relation to the hundreds of millions of vaccinations that have occurred and the protective benefits offered by COVID-19 vaccination far outweigh its risks [59,60,61,62]. Further work is required to systematically summarize the potential range of acute kidney complications following COVID-19 vaccination. All we can recommend at this point is to consider early biopsy to identify the histopathology amongst patients presenting with symptoms relating to kidney disease following vaccination. Keeping a watchful eye for patients with pre-existing immune-mediated kidney disease following vaccination may identify relapsing disease in a timely manner.

## Figures and Tables

**Figure 1 vaccines-09-01252-f001:**
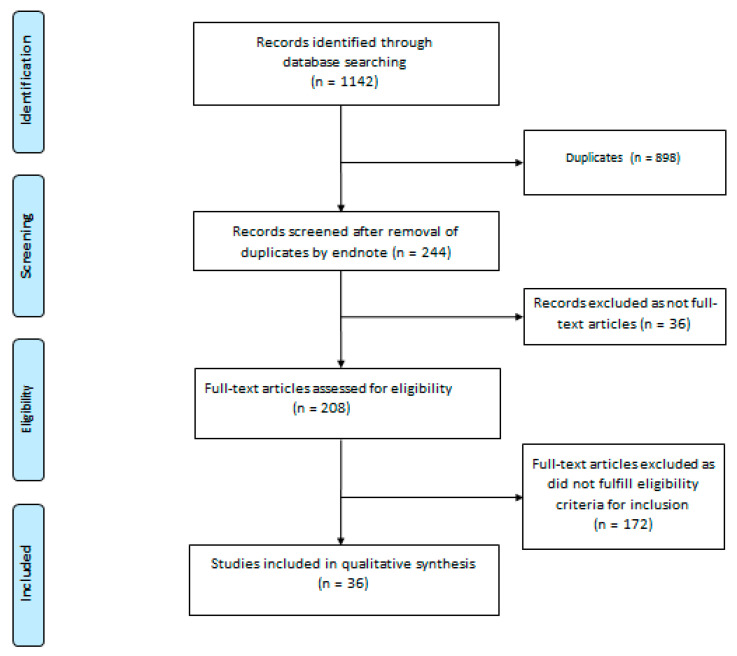
PRISMA flow diagram.

**Figure 2 vaccines-09-01252-f002:**
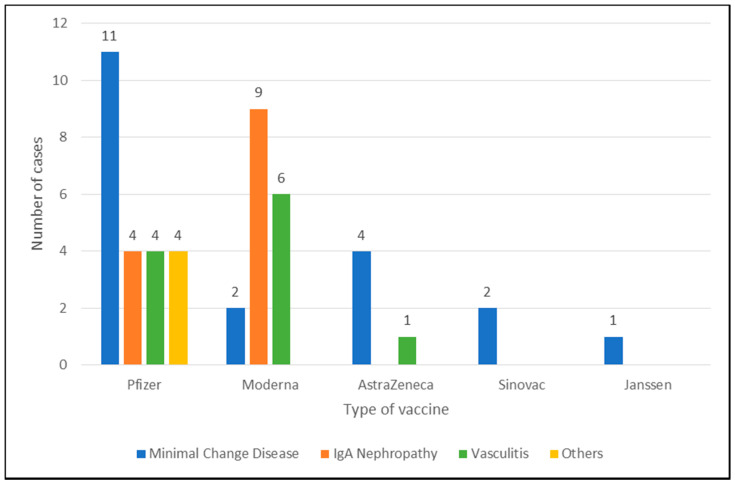
Summary of kidney pathologies and the type of COVID-19 vaccines administered.

**Table 1 vaccines-09-01252-t001:** Demographics and outcomes of patients admitted following COVID-19 vaccination with minimal change disease.

Author and Country of Case Report (Ref.)	Age (Years)	Sex	Time to Presentation from Day of Vaccination (Days)	Comorbidities	New Onset or Relapse	Vaccine Brand	Vaccine Dose	Baseline Creatinine (mg/dL)	Presentation Creatinine (mg/dL)	Presentation Proteinuria (g/Day)	Presentation Albumin (g/dL)	Hematuria	Treatment Received	Outcome
	Minimal Change Disease													
Lebedev et al., Israel [9]	50	M	10	Nil	N	Pfizer-BioNTech	1	0.78	2.31	6.9	1.93	Yes	Prednisolone	RP, DI
Maas et al., Netherlands [10]	Early 80s	M	7	Venous thromboembolisms	N	Pfizer-BioNTech	1	-	1.43	15.3	1.03	No	Prednisolone	RP, DI
D’Agati et al., United States [11]	77	M	7	T2DM, obesity, smoking, and CAD	N	Pfizer-BioNTech	1	1.0–1.3	2.33	23.2	3.0	No	Pulsed MP, prednisolone, and diuresis	RP, DI
Weijers et al., Netherlands [12]	61	F	8	Autoimmune hepatitis and hypothyroidism	N	Pfizer-BioNTech	1	-	1.47	12	2.1	No	Steroids	RP, HD received, DI on discharge
Salem et al., United States [13]	41	F	5	Asthma	N	Pfizer-BioNTech	2	-	-	14.4	2.6	Yes	-	-
Kobayashi et al., Japan [14]	75	M	7	Nil	N	Pfizer-BioNTech	2	0.96	1.24	7.72	1.1	No	Intravenous MP and prednisolone	RP, DI
Leclerc et al., Canada [15]	71	M	13	Dyslipidemia	N	Oxford- AstraZeneca	1	0.7	10.6	23.2	2.8	Yes	Pulsed MP and prednisolone	RP, HD received, DI on discharge
Anupama et al., India [16]	19	F	1	-	N	Oxford- AstraZeneca	1	-	1.09	31.8	2.15	No	Prednisolone	RP, DI
Holzworth et al., United States [17]	63	F	Less than 1 week	Hypertension and tobacco dependence	N	Moderna	1	0.7	1.48	13.4	0.7	No	Pulsed MP, prednisolone, ARB, and diuresis	RP, DI
Lim et al., South Korea [18]	51	M	28	Nil	N	Janssen	1	-	1.54	8.6	1.6	Yes	Parenteral MP and oral steroids	RP, DI
Dirim et al., Turkey [19]	65	M	17	T2DM, Hashimoto’s thyroiditis	N	Sinovac	1	-	1	11.9	1.1	No	MP and diuresis	RP, DI
Komaba et al., Japan [20]	Mid 60s	M	19	MCD	R	Pfizer-BioNTech	1	-	0.99	11.48	2.8	No	Prednisolone and cyclosporine	RP, DI
Mancianti et al., Italy [21]	39	M	3	MCD	R	Pfizer-BioNTech	1	0.9	1.8	8	2.7	No	Prednisolone	RP, DI
Kervella et al., France [22]	34	F	10 (first dose) 37 (second dose)	MCD	R	Pfizer-BioNTech	2	-	-	24 (first dose) 30 (second dose)	-	No	Steroid increased from regular dose	RP, DI
Salem et al., United States [13]	34	F	28	Steroid-sensitive MCD	R	Pfizer-BioNTech	2	-	-	12.9	2.8	No	-	-
Schwotzer et al., Switzerland [23]	22	M	3	MCD	R	Pfizer-BioNTech	1	-	0.80	-	2.3	No	Prednisolone, remained on tacrolimus	RP, DI
Morlidge et al., United Kingdom [24]	30	M	2	MCD	R	Oxford- AstraZeneca	1	-	0.93	24.1	4.7	No	Prednisolone	RP, DI
Morlidge et al., United Kingdom [24]	40	F	1	MCD	R	Oxford- AstraZeneca	1	-	-	-	-	No	Regular prednisolone, increased dose	RP, DI
Salem et al., United States [13]	33	F	21	MCD	R	Moderna	2	-	-	6.4	2.3	No	-	-

Abbreviations: ARB, angiotensin receptor blocker; CAD, coronary artery disease; DI, dialysis-independent; F, female; HD, hemodialysis; M, male; MCD, minimal change disease; MP, methylprednisolone; N, new onset; R, relapse; RP, resolution of proteinuria; and T2DM, type 2 diabetes mellitus.

**Table 2 vaccines-09-01252-t002:** Demographics and outcomes of patients admitted following COVID-19 vaccination with IgA nephropathy.

Author and Country of Case Report	Age	Sex	Time to Presentation from Day of Vaccination (Days)	Comorbidities	New Onset or Known Case	Vaccine Brand	Vaccine Dose	Baseline Creatinine (mg/dL)	Presentation Creatinine (mg/dL)	Presentation Proteinuria (g/Day)	Presentation Albumin (g/dL)	RBC per High-powered Field	Treatment Received	Hematuria Outcome
Tan et al., Singapore [25]	41	F	1	Gestational DM	N	Pfizer- BioNTech	2	-	1.73	2.03	-	>200	Pulsed MP, prednisolone, and cyclophosphamide	-
Andregg et al., Switzerland [26]	39	M	Immediately after second dose	HTN	N	Moderna	2	-	-	-	-	Macroscopic hematuria	Glucocorticoids and cyclophosphamide	RH
Abramson et al., United States [27]	30	M	1	Nil	N	Moderna	2	-	1.02	0.8	-	>30	ARB	RH
Kudose et al., United States [28]	50	F	2	HTN, obesity, and antiphospholipid syndrome	N	Moderna	2	1.3	1.7	2	-	>50	Supportive	RH
Kudose et al., United States [28]	19	M	2	Undiagnosed microhematuria for 6 months prior	N	Moderna	2	-	1.2	-	-	Gross hematuria	Supportive	RH
Park et al., United States [29]	50	M	1	CKD and HTN	N	Moderna	2	1.17	1.54	3.56	-	>50	RAASi	RH
Rahim et al., United States [30]	52	F	1	IgA nephropathy	KC	Pfizer- BioNTech	2	0.7–0.8	-	4.2	-	Gross hematuria	RAASi	RH
Perrin et al., France [31]	41	F	2	IgA nephropathy, KT patient	KC	Pfizer- BioNTech	1	-	-	0.47	-	Gross hematuria	Supportive	RH
Perrin et al., France [31]	27	F	2	IgA nephropathy, HD patient	KC	Pfizer- BioNTech	2	-	-	1.9	-	Nil gross hematuria	Supportive	RH
Perrin et al., France [31]	22	M	2 and 25 after first dose, 2 after second dose	IgA vasculitis	KC	Moderna	2	-	-	0.34	-	Nil gross hematuria	Supportive	RH
Negrea et al., United States [32]	38	F	-	IgA nephropathy	KC	Moderna	2	-	-	1.40	-	Microscopic hematuria	-	-
Negrea et al., United States [32]	38	F	-	IgA nephropathy	KC	Moderna	2	-	-	0.40	-	Microscopic hematuria	-	-
Cases with no kidney biopsy														
Park et al., United States [29]	22	F	2	Nil	N	Moderna	2	0.80	0.80	0.40	-	>50	Supportive	RH
Park et al., United States [29]	39	F	2	Nil	N	Moderna	2	-	0.80	0.90	-	>50	Supportive	RH

Abbreviations: ARB, angiotensin receptor blocker; DI, dialysis-independent; DM, diabetes mellitus; F, female; HD, hemodialysis; HTN, hypertension; M, male; MP, methylprednisolone; N, new onset; KC, known case; KT, kidney transplantation; RBC, red blood cells; RH, resolved hematuria; and T1DM, type 1 diabetes mellitus.

**Table 3 vaccines-09-01252-t003:** Demographics and outcomes of patients admitted following COVID-19 vaccination with vasculitis.

Author and Country of Case Report	Age	Sex	Time to Presentation from Day of Vaccination (Days)	Comorbidities	Vaccine Brand	Number of Vaccine Doses	Baseline Creatinine (mg/dL)	Presentation Creatinine (mg/dL)	Presentation Proteinuria (g/Day)	Presentation Albumin (g/dL)	RBC with High-Powered Field	Treatment Received	Hematuria Outcome
Anti-neutrophil Cytoplasmic Autoantibody (ANCA)-associated vasculitis													
Shakoor et al., United States [33]	78	F	28 (after first dose) and 6 (after second dose)	T2DM, HTN, and paroxysmal AF	Pfizer- BioNTech	2	0.77	1.31 (fist dose) 3.54 (second dose)	0.1	-	99 (fist dose) and 56 (second dose)	Intravenous MP and prednisolone	DI
Dube et al., United States [34]	29	F	49	Congenital diffuse cystic lung disease, awaiting lung transplant	Pfizer- BioNTech	2	0.8	1.91	0.633	4.4	12	MP, prednisolone, rituximab, and intravenous cyclophosphamide	DI
Sekar et al., United States [35]	52	M	14	HTN	Moderna	2	1.11	8.41	-	-	Microscopic hematuria	Prednisolone, rituximab, and cyclophosphamide	DD
Andregg et al., Switzerland [26]	81	M	-	Nil	Moderna	1	-	-	-	-	-	High-dose steroids, cyclophosphamide, and plasma exchange	DI
Granulomatous Vasculitis													
Gillion et al., Belgium [36]	77	M	28	Nil significant	Oxford- AstraZeneca	1	1.2	2.7	0.07	-	No hematuria	MP	DI
IgA Vasculitis (Henoch–Schonlein Purpura)													
Obeid et al., Switzerland [37]	78	F	7	IgA vasculitis with leukocytoclastic vasculitis	Moderna	1	1.08	1.18	-	-	150	MP	DI
Park et al., United States [29]	67	M	28	CKD and HTN	Moderna	1	1.20	2.90	2.10	-	>50	Steroid	DI
Anti-glomerular Basement Membrane (Anti-GBM) Disease													
Tan et al., Singapore [25]	60	F	1	Hyperlipidemia	Pfizer- BioNTech	2	-	6.11	7.58	-	>200	Pulsed MP, prednisolone, cyclophosphamide, and plasma exchange	-
Sacker et al., United States [38]	‘Older Woman’	F	14	Nil significant	Moderna	2	-	7.8	1.9	-	Gross hematuria	MP, cyclophosphamide, plasma exchange, and HD	DD
Lupus Nephritis													
Tuschen et al., Germany [39]	42	F	7	SLE with lupus nephritis class V	Pfizer- BioNTech	1	-	-	6	-	No hematuria	MMFand prednisolone	DI

Abbreviations: AF, atrial fibrillation; CKD, chronic kidney disease; DD, dialysis-dependent; DI, dialysis-independent; F, female; HD, hemodialysis; HTN, hypertension; M, male; MMF, mycophenolate mofetil; MP, methylprednisolone; SLE, systemic lupus erythematous; and T2DM, type 2 diabetes mellitus.

## Data Availability

Not applicable.

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
