# Peer review of "New-Onset and Relapsed Kidney Histopathology Following COVID-19 Vaccination: A Systematic Review"

_vaccines, 2021, doi:10.3390/vaccines9111252_

Round 1
Reviewer 1 Report
Thank you for your valuable work.
I think with Covid pandemic, with regard to the side effects of vaccine, we need meticulous and accurate reviews.
Author Response
Thank you for your review and comments.
Reviewer 2 Report
The authors present a thorough review of publications that reported the presence of kidney glomerulopathy manifested shortly after receiving the first or both doses of the Covid 19 vaccine. Independently of my appreciation of the paper, I have some remarks and comments.
- The authors stated in the conclusion that the frequency of kidney diseases after Covid 19 vaccination is extremely small. This statement should be supported by some data. At first, there is a rather high dropout rate in the papers primarily chosen. This may suggest that the real frequency of kidney inflammatory complications might be higher, especially, not necessarily all cases were reported in the literature. I believe that vaccination complications reported under the umbrella of vasculitis diagnosis may not be as extremely small as the Authors believe. The paper is calling for attention to kidney complications after vaccination and for starting a systematic study on that.
- In the paper mRNA vaccines are frequently quoted, does it mean anything.
- The medical action taken may reflect the severity of the complication reported. Therefore, it might be good to provide information about the treatment in a comprehensive way. This is shown on a single case basis but summarizing the treatment will be impressive.
- Histopathology is very important, it might be also useful to summarize histopathology results showing also the grading.
Some paragraphs might be less wordy .
Author Response
Comment 1: The authors stated in the conclusion that the frequency of kidney diseases after Covid 19 vaccination is extremely small. This statement should be supported by some data. At first, there is a rather high dropout rate in the papers primarily chosen. This may suggest that the real frequency of kidney inflammatory complications might be higher, especially, not necessarily all cases were reported in the literature. I believe that vaccination complications reported under the umbrella of vasculitis diagnosis may not be as extremely small as the Authors believe. The paper is calling for attention to kidney complications after vaccination and for starting a systematic study on that.
Thank you for your comment. We would like to stress that our systematic search is focused on ‘kidney histopathology’ - demonstrated from the search terms in our criteria of article selection, as opposed to identifying all forms of renal manifestations following COVID-19 vaccination (which would be far more common given the multidimensional nature in which kidney impairment can occur as a secondary complication of various systemic diseases - especially the likelihood of AKI with systemic illness). We have added additional points in p. 8 and p. 9 (final paragraph of the discussion section and conclusion) to address this. Additional references (reference 59 to 62) of findings from recent randomized-controlled trials are inserted to support our statement that the protective benefits offered by COVID-19 vaccination far outweigh its risks, despite the kidney histopathologies reported alongside other potential renal manifestations. We have also modified the title of our article to clarify the scope of our systematic review.
Comment 2: In the paper mRNA vaccines are frequently quoted, does it mean anything.
Thank you for your comment. In the paper, the term ‘mRNA vaccines’ is used as an umbrella term to discuss the COVID-19 vaccines with vaccine technologies based on mRNA (such as Pfizer-BioNTech and Moderna). We have reviewed the manuscript and modified our description of these vaccines where it is unclear.
Comment 3: The medical action taken may reflect the severity of the complication reported. Therefore, it might be good to provide information about the treatment in a comprehensive way. This is shown on a single case basis but summarizing the treatment will be impressive.
Thank you for your comment. The conventional treatment regime for each kidney histopathology that we describe does differ and it is a fact of real world practice that individual cases were treated differently depending upon local policy. We have provided information on the treatment received for each patient in Tables 1-3 (at the end of the manuscript) for Minimal Change Disease, IgA Nephropathy and the Vasculitis groups. The manuscript text summarizes the specific treatments received amongst patients in these respective groups, where there is more frequent occurrence of histopathology following COVID-19 vaccination. Due to the different histopathological presentations observed in the ‘Other Cases’, it was important to address each case individually instead of grouping them altogether.
Comment 4: Histopathology is very important; it might be also useful to summarize histopathology results showing also the grading.
Thank you. We have summarized the grading of histopathology (i.e. lupus nephritis case in p. 5 and p. 6) in our manuscript where this is presented from the original case report, to demonstrate the severity of disease.
Comment 5: Some paragraphs might be less wordy.
We have reviewed our manuscript again to ensure our main text is more succinct.
Reviewer 3 Report
Unfortunately, my manuscript copy did not carry the Tables so I cannot fully assess the soundness of the results. The review seems to adhere to high standards, perhaps it should describe more the age-dependency of the kidney damages and report more on the prevalence of such diseases in the countries where they have been observed. It is without reference to the final comment of the discussion, why point to AstraZeneca which has been used in hundreds of millions of doses in Western countries and not underreported vaccines used mainly in East Asian countries? The underreporting is a weakness of this review and it should be underlined throughout and in the abstract. Some typos should be corrected (see p. 7 "noted increased production of ANCA production" ).
Author Response
Comment 1: Unfortunately, my manuscript copy did not carry the Tables so I cannot fully assess the soundness of the results.
Thank you for letting us know. We have included Tables 1-3 at the end of the manuscript for your reference.
Comment 2: The review seems to adhere to high standards, perhaps it should describe more the age-dependency of the kidney damages and report more on the prevalence of such diseases in the countries where they have been observed.
Thanks. This is an important point that we fully acknowledge. As this systematic review aimed to capture all of the published case reports on COVID-19 vaccine associated kidney histopathologies (case reports and series on this topic that were available within current literature), we feel this would not be a fair reflection on the prevalence of these conditions within the listed countries.
Comment 3: It is without reference to the final comment of the discussion, why point to AstraZeneca which has been used in hundreds of millions of doses in Western countries and not underreported vaccines used mainly in East Asian countries? The underreporting is a weakness of this review and it should be underlined throughout and in the abstract.
Thank you. We have now reworded the final paragraph prior to the conclusion (p.8) to make this point clear. We have now removed the hypothetical statement on under-reporting.
Comment 4: Some typos should be corrected (see p. 7 "noted increased production of ANCA production").
We have now carefully examined the manuscript for any typo errors and made corrections as indicated.
Reviewer 4 Report
Strength: in this review, thirty-six articles about kidney pathologies that may have followed COVID-19 vaccination were included in this systematic while the principal pathology encountered was Minimal change disease, followed by IgA Nephropathy (14 cases) and Vasculitis (10 cases). Other cases include relapse of membranous nephropathy, acute rejection of kidney transplant, relapse of IgG4 nephritis, new-onset renal thrombotic microangiopathy and scleroderma renal crisis following COVID-19 vaccination. There was no mortality reported in any of the included cases. Patients in all but one case largely recovered and did not require long-term renal replacement therapy. the manuscript is well organized and interesting for readers. however:
Weakness: please check carefully english language.
It would be good to add some figures or schemes about the different diseases to make it more attractive visually
Minimal change disease, followed by IgA Nephropathy (14 cases) and Vasculitis (10 cases).
Author Response
Comment 1: Weakness: please check carefully english language.
Thanks. We have now carefully examined the manuscript for any language and typo errors, which are now corrected.
Comment 2: It would be good to add some figures or schemes about the different diseases to make it more attractive visually. Minimal change disease followed by IgA Nephropathy (14 cases) and Vasculitis (10 cases).
Thank you for your remark. We have now inserted Figure 2 to make our presentation more visually attractive.